# Sulfur-Doped Organosilica Nanodots as a Universal Sensor for Ultrafast Live/Dead Cell Discrimination

**DOI:** 10.3390/bios12111000

**Published:** 2022-11-10

**Authors:** Yan-Hong Li, Jia Zeng, Zihao Wang, Tian-Yu Wang, Shun-Yu Wu, Xiao-Yu Zhu, Xinping Zhang, Bai-Hui Shan, Cheng-Zhe Gao, Shi-Hao Wang, Fu-Gen Wu

**Affiliations:** State Key Laboratory of Bioelectronics, School of Biological Science and Medical Engineering, Southeast University, 2 Sipailou Road, Nanjing 210096, China

**Keywords:** nanoprobe, fluorescence imaging, live/dead staining, bacteria, mammalian cells, fungi

## Abstract

Rapid and accurate differentiation between live and dead cells is highly desirable for the evaluation of cell viability. Here, we report the application of the orange-emitting sulfur-doped organosilica nanodots (S-OSiNDs) for ultrafast (30 s), ultrasensitive (1 μg/mL), and universal staining of the dead bacterial, fungal, and mammalian cells but not the live ones, which satisfies the requirements of a fluorescent probe that can specifically stain the dead cells. We further verify that the fluorescence distribution range of S-OSiNDs (which are distributed in cytoplasm and nucleus) is much larger than that of the commercial dead/fixed cell/tissue staining dye RedDot2 (which is distributed in the nucleus) in terms of dead mammalian cell staining, indicating that S-OSiNDs possess a better staining effect of dead cells than RedDot2. Overall, S-OSiNDs can be used as a robust fluorescent probe for ultrafast and accurate discrimination between dead and live cells at a single cell level, which may find a variety of applications in the biomedical field.

## 1. Introduction

Rapidly, sensitively, and accurately distinguishing between live and dead cells is of tremendous significance to various biomedical fields, such as cancer therapy and microbial infection treatment [1,2,3,4,5,6,7]. However, a simple and efficient method for fast and accurate live/dead cell discrimination at a single-cell level is still lacking, and additionally, some existing methods, such as atomic force microscopy [8], Fourier transform infrared spectroscopy [9], electron microscopy [10,11], Raman spectroscopy [12,13], and nucleic acid sequence-based amplification [14], are complicated, expensive, time-consuming, and laborious, seriously restricting their practical applications in the discrimination between live and dead cells.

In recent years, fluorescence labeling technology has gained much attention due to its outstanding advantages, including rapid response, simple operation, high sensitivity, ease of quantification, etc. [15,16,17,18,19,20,21]. The frequently used fluorescent reagents capable of realizing the discrimination between live and dead cells include SYTOX Green nucleic acid stain [22,23], propidium iodide (PI) [24], rhodamine 123 [25], fluorescein derivatives [26,27,28], calcein acetoxymethyl ester [29], RedDot2 [30], and carbon dots [31,32,33,34,35,36,37,38]. Most of these reagents selectively stain the dead cells via penetrating compromised cell membranes and staining the cell nucleus (e.g., RedDot2). However, these commercial dyes are relatively expensive, sometimes toxic, and require a long staining time before imaging. Carbon dots (a class of alternative materials for staining dead cells) usually have multicolor fluorescence emission, which overlaps with that of the live cell probes and severely limits their practical applications. Therefore, developing economical, low-toxic, and rapid imaging reagents with excellent optical properties for the discrimination between live and dead cells is highly required.

Fluorescent silicon nanoparticles (SiNPs), with satisfactory biocompatibility, outstanding photoluminescence stability, and convenient surface modification property [39], have been extensively applied in organic solar cells [40], antibacterial application [41,42], drug delivery [43], cancer therapy [44,45], sensing [46,47,48,49,50,51], and bioimaging [3,52,53,54]. Organosilica nanodots (OSiNDs) are one typical type of SiNPs prepared from organic silane molecules. In a previous research, our group fabricated OSiNDs through the hydrothermal treatment of rose bengal (RB) and 3-[2-(2-aminoethylamino)ethylamino]propyl-trimethoxysilane, and the OSiNDs possess the properties of small size (2.0 nm) and green fluorescence emission (525 nm) [54]. The OSiNDs were applied as an imaging reagent for lysosomal imaging for various types of cells and cells in different states (such as living, fixed, and permeabilized cells). Besides, in a later study, the OSiNDs were also applied for the wash-free, rapid, and universal staining of dead mammalian, bacterial, and fungal cells [3]. In addition, our group prepared another type of OSiNDs via a one-step hydrothermal reaction of an epoxy group-containing silane molecule, 3-glycidoxypropyltrimethoxysilane (GPTMS), and RB, and the resulting OSiNDs possessed the properties of small size (3.7 nm) and green fluorescence emission (529 nm) [42]. The OSiNDs were applied as an imaging reagent for visualizing various bacteria/biofilms. However, the above-mentioned OSiNDs all possess relatively short emission wavelengths (green fluorescence), which may decrease the imaging accuracy and limit further applications in organs or tissues in vivo. Besides, up till now, there is only one kind of OSiNDs that can be used for the fluorescence imaging-based discrimination between live and dead cells [3]. Therefore, the development of a sensitive and rapid fluorescent probe with long emission wavelengths (orange, red, or near-infrared fluorescence) for live/dead cell discrimination is highly desirable.

In 2021, we prepared orange-emitting sulfur-doped OSiNDs (S-OSiNDs) with a photoluminescence quantum yield (PLQY) of 13.4% (solvent: water) by the solvothermal treatment of citric acid, urea, and bis[3-(triethoxysilyl)propyl]tetrasulfide in *N*, *N*-dimethylformamide (DMF) at 200 °C for 12 h [51]. The S-OSiNDs realized the detection of multiple metal ions and achieved cancer/normal cell discrimination. Besides, the metal ion-induced fluorescence quenching of S-OSiNDs could be selectively restored by glutathione (GSH), and thus the metal ion-treated S-OSiNDs exhibited a sensitive GSH detection capability. Here, we demonstrate that the same S-OSiNDs can be used as a fluorescent probe for the differentiation between live and dead microbial (bacterial and fungal) and mammalian cells (Figure 1). The S-OSiNDs can realize ultrafast (30 s), highly sensitive (required dose of S-OSiNDs: 1 μg/mL), accurate, universal, and selective staining of dead cells. Compared with RedDot2 (a commercial far-red cell membrane-impermeant nuclear dye suitable for selective dead/fixed cell/tissue staining), the fluorescence distribution range of S-OSiNDs (which are distributed in cytoplasm and nucleus) is much larger than that of RedDot2 (which is distributed in the nucleus) in terms of dead mammalian cell staining. The results indicate that S-OSiNDs possess a better staining effect of dead cells than RedDot2 and represent a promising probe for accurate discrimination between live and dead cells.

## 2. Materials and Methods

### 2.1. Preparation of Live/Dead Cells

For bacterial (*Escherichia coli* (*E*. *coli*) and *Staphylococcus aureus* (*S*. *aureus*)) and fungal cells (*Saccharomyces cerevisiae* (*S*. *cerevisiae*) yeast), the cells (10^8^ colonies forming unit (CFU)/mL) were centrifugated (8000 rpm, 5 min) and then collected after being washed with physiological saline. Then, the cells were divided into two groups (live and dead groups). The cells in the dead group were treated with 1% benzalkonium bromide solution for 2 h, followed by washing with physiological saline. The cells in the live group were resuspended in physiological saline. For mammalian cells (HPAEpiCs (normal human pulmonary alveolar epithelial cells) and A549 (human lung cancer cells)), the cells were cultured in a 96-well glass bottom plate (5 × 10^3^ cells/well). Next, the dead cells were obtained after treating the cells with ethanol for 10 min, while the live cells were incubated in Dulbecco’s modified Eagle’s medium (DMEM).

### 2.2. Evaluation of the Staining Performance of S-OSiNDs toward Live and Dead Bacteria

Live/dead *E*. *coli and S*. *aureus* cells (10^8^ CFU/mL) were separately treated with different concentrations of S-OSiNDs (0, 1, 2, 5, 10, 20, and 50 μg/mL) for different time periods (30 s, 1, 2, 5, 10, and 30 min). Then, the cells were washed with physiological saline for 3 times and evaluated by flow cytometry using a flow cytometer (NovoCyte 2070R, ACEA Biosciences Inc., San Diego, CA, USA). In addition, the treated cells were imaged under a confocal microscope (TCS SP8, Leica, Wetzlar, Germany) at an excitation wavelength of 552 nm.

### 2.3. Evaluation of the Staining Performance of S-OSiNDs toward Live and Dead Fungi

For confocal imaging, the live/dead *S*. *cerevisiae* cells (10^8^ CFU/mL) were exposed to different concentrations of S-OSiNDs (1, 2, 5, 10, 20, and 50 μg/mL) for different time periods (30 s, 1, 2, 5, 10, and 30 min), and then imaged using the confocal microscope at an excitation wavelength of 552 nm. For flow cytometry analysis, the live/dead *S*. *cerevisiae* cells (10^8^ CFU/mL) were first exposed to different concentrations of S-OSiNDs (0, 1, 2, 5, 10, 20, and 50 μg/mL) for different time periods (30 s, 1, 2, 5, 10, and 30 min). Then, after washing the treated fungal cells with physiological saline for 3 times, we quantified the cellular fluorescence intensities by flow cytometry.

### 2.4. Evaluation of the Staining Performance of S-OSiNDs toward Normal and Cancerous Mammalian Cells

For confocal imaging, the normal/cancerous mammalian cells (HPAEpiC/A549) were mixed with different concentrations of S-OSiNDs (1, 2, 5, 10, 20, or 50 μg/mL) for different time periods (30 s, 1, 2, 5, 10, or 30 min). Afterward, the cells were imaged under the confocal microscope at an excitation wavelength of 552 nm. For flow cytometry analysis, the live/dead HPAEpiCs and A549 cells were first treated with different concentrations of S-OSiNDs (0, 1, 2, 5, 10, 20, or 50 μg/mL) for different time periods (30 s, 1, 2, 5, 10, or 30 min). Then, after washing the treated HPAEpiCs and A549 cells with phosphate-buffered saline (PBS, pH 7.4) for 3 times, we measured the cellular fluorescence intensities using flow cytometry.

### 2.5. Comparison between S-OSiNDs and RedDot2 on the Live/Dead Cell Discrimination Performance

To compare the live/dead differentiation performance of S-OSiNDs and RedDot2 (a commercial dye for nuclear DNA imaging), the live/dead bacteria (*E*. *coli*/*S*. *aureus*), fungi (*S*. *cerevisiae* yeast), and normal/cancerous mammalian cells (HPAEpiC/A549) were stained by the mixture of S-OSiNDs (5 μg/mL) and RedDot2 (diluted with physiological saline (for bacteria and fungi)/PBS (for mammalian cells) using a 1:200 ratio) for 10 min in the dark. Then the cells were imaged by the confocal microscope. The excitation wavelength of S-OSiNDs was 552 nm, while that of RedDot2 was 638 nm. Since the fluorescence emission color of S-OSiNDs was orange, which might be interfered with that of RedDot2 (red), the imaging color of S-OSiNDs was set as green.

## 3. Results and Discussion

### 3.1. Staining Performance of S-OSiNDs for Live and Dead Bacteria

S-OSiNDs were prepared according to our previous report [51]. The as-prepared S-OSiNDs exhibit a uniform size distribution with an average size of 1.3 ± 0.3 nm (Appendix A). The ultraviolet–visible (UV–vis) absorption spectrum of the S-OSiNDs shows an absorption peak at 335 nm and two broad absorption bands centering at around 470 and 558 nm, the latter of which is related to the fluorescence excitation of S-OSiNDs (Appendix A). As shown by the fluorescence emission spectra, S-OSiNDs display excitation-independent fluorescence emission with the maximum excitation and emission peaks at 558 and 583 nm, respectively (Appendix A). Besides, the PLQY of S-OSiNDs in water was calculated to be 13.4%. The Fourier transform infrared spectroscopy (FTIR) and X-ray photoelectron spectroscopy (XPS) data demonstrated that the obtained S-OSiNDs possess various groups/moieties, including C–H, O–H/N–H, C–O/C–N, C–C, Si–O, C═O, C–S, etc. (Appendix A).

To evaluate if S-OSiNDs can be adapted to distinguish between live and dead bacterial cells, we chose *E*. *coli* and *S*. *aureus* as the representatives of the Gram-negative and Gram-positive bacteria, respectively. The results in Figure 1A and Figure 2A revealed that the live bacterial cells (both *E*. *coli* and *S*. *aureus*) were hardly labeled by all the tested concentrations of S-OSiNDs, while the dead cells were observed to exhibit strong fluorescence signals. According to the flow cytometric results (Figure 1B–D and Figure 2B–D), the dead cells exhibited much higher fluorescence intensities than the live ones. Also, the effect of staining time on the live/dead cell discrimination performance of S-OSiNDs was further evaluated (Figure 3 and Figure 4). Strong fluorescence signals were observed for the dead cells after incubation with S-OSiNDs after 30 s, whereas the live cells that we treated with S-OSiNDs for different time periods did not display noticeable fluorescence. Moreover, the fluorescence intensities of the dead cells reached a plateau after 1 min (for *E*. *coli*)/30 s (for *S*. *aureus*) (Figure 3D and Figure 4D), demonstrating that the dead bacterial cell staining by S-OSiNDs is very fast. Besides, the results of the cellular fluorescence intensities obtained by flow cytometry (Figure 3B–D and Figure 4B–D) agreed well with the confocal imaging results (Figure 3A and Figure 4A).

Furthermore, we compared the fluorescence intensities between the S-OSiND-stained *E*. *coli* and *S*. *aureus*. According to the flow cytometric results (Figure 1D, Figure 2D, Figure 3D and Figure 4D), the fluorescence intensities of the dead *E*. *coli* cells were higher than those of the dead *S*. *aureus* cells at the same S-OSiND concentrations/incubation time (at the plateau stages). This is because the size/volume of *E*. *coli* is larger than that of *S*. *aureus*, and therefore a higher intracellular content of S-OSiNDs can be found in the dead *E*. *coli* cells than that in the dead *S*. *aureus* cells.

Collectively, the above results demonstrated that S-OSiNDs were able to achieve rapid and accurate differentiation between live and dead bacterial cells.

### 3.2. Staining Performance of S-OSiNDs for Live and Dead Fungi

Besides the bacterial cells, we also tested the capacity S-OSiNDs to distinguish between live and dead fungal cells. To this end, the *S*. *cerevisiae* yeast was chosen. As shown in Figure 5A, only dead yeasts were successfully stained by the synthesized S-OSiNDs with strong red fluorescence, while the live yeasts did not show any fluorescence (compared with the untreated control group). In addition, the fluorescence intensities of the dead and live yeasts were measured using flow cytometry. The results showed that at all the tested concentrations, the cellular content of S-OSiNDs in the dead yeasts was much higher than that in the live cells (Figure 5B–D). Moreover, we further investigated the effect of incubation time on the live/dead cell discrimination performance of S-OSiNDs. It could be found that strong red fluorescence signals could be seen in the dead cells after incubation with S-OSiNDs for 30 s or longer, whereas the live cells that were treated with S-OSiNDs for different time periods did not display noticeable fluorescence (Figure 6A). In addition, as revealed by the corresponding flow cytometric results (Figure 6B–D), the fluorescence intensity of the dead yeasts reached a plateau after 30 s, demonstrating the rapid S-OSiND staining effect toward dead yeasts.

To check the sensitivity of live/dead fungal cell discrimination of S-OSiNDs, we first defined the discrimination sensitivity as *I*_dead_/*I*_live_ (in which *I*_dead_ represents the fluorescence intensity of dead cells, and *I*_live_ represents the fluorescence intensity of live cells). When the value of *I*_dead_/*I*_live_ is above 1, the live and dead cells can be theoretically discriminated. In addition, the larger the value of *I*_dead_/*I*_live_ is, the better the live/dead cell discrimination effect is. As shown in Figure 6D, after staining for 30 s, we could see that the value of *I*_dead_/*I*_live_ was 15.3–16.8, which is much larger than 1, indicating that the discrimination is successful. In addition, Figure 5D exhibited that the fluorescence intensity of dead cells was concentration-dependent, and a higher S-OSiND concentration resulted in a stronger fluorescence emission, which accordingly led to a higher *I*_dead_/*I*_live_ value and a higher discrimination sensitivity.

Collectively, the above results suggested that S-OSiNDs could achieve fast and accurate live/dead fungal cell discrimination.

### 3.3. Staining Performance of S-OSiNDs for Normal and Cancerous Mammalian Cells

Inspired by the above-mentioned live/dead microbial cell discrimination results of S-OSiNDs, we further tested the feasibility of using S-OSiNDs for distinguishing between live/dead normal/cancerous mammalian cells. We chose the normal HPAEpiCs and the cancerous A549 cells as two representative mammalian cell lines for the live/dead cell staining assay. As shown in Figure 7 and Figure 8, the dead HPAEpiCs and A549 cells could be selectively labeled by different concentrations of S-OSiNDs, whereas almost no fluorescence was detected from the live cells even when the concentration increased to 50 μg/mL (Figure 7A and Figure 8A). The corresponding flow cytometric results in Figure 7B–D and Figure 8B–D further indicated that in all cases, the dead cells exhibited much higher fluorescence intensities than the live cells. The fluorescence intensity of the dead cells was concentration-dependent, and a dramatically enhanced fluorescence could be observed even at a very low S-OSiND concentration of 1 μg/mL.

Moreover, we checked the effect of staining time on the live/dead mammalian cell differentiation performance of S-OSiNDs. As shown in Figure 9A and Figure 10A, after incubation for only 30 s, the dead cells could be successfully stained by S-OSiNDs with strong red fluorescence, and the red fluorescence signals were distributed in the entire cells, including cytoplasm and nucleus, whereas no fluorescence was detected in the live cells even when the staining time was increased to 30 min. The results of the cellular fluorescence intensities obtained by flow cytometry (Figure 9B–D and Figure 10B–D) were consistent with the confocal imaging results. Additionally, according to the flow cytometric results in Figure 9B–D and Figure 10B–D, we could clearly see that there were no evident changes in the fluorescence intensity of the dead HPAEpiCs and A549 cells with the increase of staining time after 30 s.

Furthermore, we compared the fluorescence intensities between the HPAEpiCs and A549 cells. It was found that the fluorescence intensities of the dead HPAEpiCs were higher than those of the dead A549 cells at the same S-OSiND concentrations/incubation time (at the plateau stages) (Figure 7D, Figure 8D, Figure 9D and Figure 10D). Besides, according to the flow cytometric results (data not shown), the size/volume of HPAEpiCs is larger than that of A549 cells. Therefore, more intracellular S-OSiNDs can be accommodated in the dead HPAEpiCs than in the dead A549 cells. 

Collectively, S-OSiNDs represent a promising and universal fluorescent probe for successfully discriminating between live and dead cells and realizing ultrafast (30 s) and sensitive staining (required dose of S-OSiNDs: 1 μg/mL) of dead cells.

### 3.4. Comparison between S-OSiNDs and RedDot2 on the Discrimination between Live and Dead Cells

To further evaluate the capability of S-OSiNDs in live/dead cell discrimination and uncover the selective labeling mechanism of dead cells, we treated the live and dead cells with the mixtures of S-OSiNDs and RedDot2. As shown in Figure 11, little fluorescence was observed in the live cells, revealing that the live cells were neither stained by S-OSiNDs nor RedDot2. On the other hand, strong yellow fluorescence could be observed in the merged confocal images of dead cells, confirming that S-OSiNDs (shown as green in the images) and RedDot2 (red fluorescence) can both selectively light up the dead cells and the dead cell staining mechanism of S-OSiNDs is the same as that of RedDot2. Specifically, for the dead HPAEpiCs and A549 cells, the red fluorescence of RedDot2 was mainly located in the nucleus, while the green signals (pseudo color) of S-OSiNDs were distributed in both the cytoplasm and nucleus, which illustrated that the fluorescence distribution range of S-OSiNDs was much larger than that of RedDot2 under the same staining condition. The larger fluorescence distribution range of S-OSiNDs indicated their better staining ability of the dead cells than that of RedDot2, which may help them to realize more sensitive and accurate discrimination of dead and live cells.

### 3.5. Cytotoxicity Evaluation of S-OSiNDs

Next, the cytotoxicity of S-OSiNDs to the bacterial, fungal, and mammalian cells was also evaluated. The corresponding results are shown in Figure 12. It could be found that S-OSiNDs had negligible toxicity to *E*. *coli*, *S*. *aureus*, and yeast cells at concentrations of 50 μg/mL. Even when the concentration increased to 500 μg/mL, S-OSiNDs only had slight toxicity to *E*. *coli* (viability: ~76%), *S*. *aureus* (viability: ~84%), and yeast (viability: ~85%). When the concentration of S-OSiNDs was 500 µg/mL, the relative viabilities of HPAEpiCs and A549 cells were ~93% and ~99%, respectively. It is worth noting that the S-OSiND concentration of 500 μg/mL is much higher than the working concentration of 5 μg/mL. These results revealed the superb cytocompatibility of S-OSiNDs, ensuring their practical applications in cell imaging.

## 4. Conclusions

In this work, we demonstrated that S-OSiNDs could realize sensitive (1 μg/mL), ultrafast (30 s), and selective fluorescence staining of dead bacterial, fungal, and mammalian cells. We have also proved that S-OSiNDs possessed good cytocompatibility. As a result, S-OSiNDs can be adopted as a robust fluorescent probe for successful fluorescence discrimination between live and dead cells regardless of the cell type. For mammalian cells, we confirmed that the fluorescence distribution range of S-OSiNDs (which are distributed in the cytoplasm and nucleus) is much larger than that of RedDot2 (which is distributed in the nucleus). These advantages demonstrate the bright application prospect of S-OSiNDs in cell imaging and cell viability evaluation. We surely believe that the S-OSiNDs can find a variety of applications in the biomedical field.

## Data Availability

Not applicable.

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
