# Peer review of "Sulfur-Doped Organosilica Nanodots as a Universal Sensor for Ultrafast Live/Dead Cell Discrimination"

_biosensors, 2022, doi:10.3390/bios12111000_

Round 1

Reviewer 1 Report

This paper describes a topic on the rapid and accurate differentiation between live and dead cells is highly desirable for the evaluation of cell viability. The results show that the applications of the orange-emitting sulfur-doped OSiNDs (S-OSiNDs) for wash-free, ultrafast (30 s, much shorter than the staining time, 10 min, of RedDot2, a commercial dye for selective dead cell staining), ultrasensitive (1 μg/mL), and universal staining of the dead bacterial, fungal, and mammalian cells but not the live ones, which satisfies the requirements of a fluorescent probe that can specifically stain the dead cells. The fluorescence distribution range of S-OSiNDs (which are distributed in cytoplasm and nucleus) is much larger than that of RedDot2 (which is distributed in nucleus) in terms of the discrimination between live and dead mammalian cells, indicating that S-OSiNDs possess a better staining ability of dead cells than RedDot2. Overall, S-OSiNDs can be used as a robust fluorescent probe for ultrafast and accurate discrimination between dead and live cells at a single cell level, which may find a variety of applications in the biomedical field.

Meanwhile, the following comments should be addressed before publications.

1.      It is strongly recommended that the authors should mention clearly the newly developed and /or found point of in section introduction, compared with papers already reported in this field.

2.      The authors present the results of orange-emitting sulfur-doped OSiNDs (S-OSiNDs) methods are currently widely used, but the technical and academic descriptions are still deficient. The authors should provide more technical and academic descriptions on what different/ effect compared with existed S-OSiNDs works.

3.      The authors should compare clearly what the difference for the sensor for fluorescence labeling technology is installed, how to improve efficiency for accurate discrimination between live and dead cells?

4.      The S-OSiNDs could achieve fast and accurate live/dead fungal cell discrimination, and how to define and change the sensitivity/ discrimination?

5. How to confirm and avoid the S-OSiNDs generate any effect between live and dead cells?

6.      The authors should show and compare the quantitative data of discrimination.

Adding the references from 2021 to 2022 is recommended.

Author Response

Dear respected reviewer,

We have attached a separate file to address your very helpful and constructive comments and suggestions! Many thanks!

Reviewer 2 Report

The manuscript "Sulfur-Doped Organosilica Nanodots as a Universal Sensor for Ultrafast Live/Dead Cell Discrimination" by Wu and coworkers present an interesting results using S-OSiNDs.

The following comments need to be addressed before publication in Biosensors

-        For evaluation the effect of incubation time, why the 5 ug/L is used

-        In Figures 1-10, add dead cells and live cells on Figures B and C

-        Comparison between the cytometric results of both E. coli and S. aureus cells  need to be discussed

-        Same thing for HPAEp-234 iCs  and  A549 cells 

Author Response

(The authors gave the same response as above.)

Round 2

Reviewer 1 Report

I think that the revised manuscript could be accepted.